# Dataset Generation Patterns for Evaluating Knowledge Graph Construction

Markus Schröder, Christian Jilek, and Andreas Dengel

[1] Smart Data & Knowledge Services Dept., DFKI GmbH, Kaiserslautern, Germany
[2] Computer Science Dept., TU Kaiserslautern, Germany
{markus.schroeder, christian.jilek, andreas.dengel}@dfki.de

**Abstract.** Confidentiality hinders the publication of authentic, labeled datasets of personal and enterprise data, although they could be useful for evaluating knowledge graph construction approaches in industrial scenarios. Therefore, our plan is to synthetically generate such data in a way that it appears as authentic as possible. Based on our assumption that knowledge workers have certain habits when they produce or manage data, generation patterns could be discovered which can be utilized by data generators to imitate real datasets. In this paper, we initially derived 11 distinct patterns found in real spreadsheets from industry and demonstrate a suitable generator called Data Sprout that is able to reproduce them. We describe how the generator produces spreadsheets in general and what altering effects the implemented patterns have.

**Keywords:** Pattern Language · Generator · Synthetic Data

## 1 Introduction and Motivation

Personal and enterprise data is usually produced and managed by knowledge workers during their work. One of our research efforts[3] is concerned with the construction of enterprise knowledge graphs from such datasets. Although, in the past we worked with various data assets, we were not allowed to publish them (together with labeled data) because of usual confidentiality reasons in personal and enterprise data. Even if this would be possible under certain circumstances, there is still a high effort to label such data with intended knowledge graphs to conduct meaningful evaluations. Therefore, our plan is to synthetically generate datasets in such a way that they appear as authentic as possible.

When knowledge workers work with data, we observe that they show certain behaviors. We assume that whenever data is entered or modified, a user tends to do it in a way the person is used to. This also includes habits or workarounds which may result in rather messy datasets, especially, if data management strategies are neglected. A particular way in which something is done or organized is usually called a *pattern*. By exploring enterprise data and interviewing its users, such reoccurring patterns may be collected and cataloged.

---

[3] https://comem.ai/SensAI

This emerging pattern language could then be utilized by data generators that reproduce these patterns to imitate real datasets. Such synthetically generated data, if authentic enough, would appear like knowledge workers had produced them in the first place. By mixing patterns in various ways, generators are able to produce arbitrary large and complex data assets – even with pattern combinations which have not been observed before. That is why the construction of knowledge graphs from such data can easily become a non-trivial task: since patterns and especially their combinations will typically introduce ambiguities, a simple pattern recognition procedure may not be sufficient enough.

Our envisioned generators take two inputs: the generation patterns and a given knowledge graph. Since generators know what statements resulted in what data, the provenance information can be used to measure the performance of construction approaches on that data. Researchers are also able to generate synthetic datasets that matches just those patterns present in their non-publishable real-world ones. Computational results are expected to be on the same level for both the real and the synthetic datasets (due to identical complexity as expressed by the underlying patterns), thus making the synthetic-based results valid.

Generation patterns are usually dependent on a given domain (such as chemistry, biology, healthcare) and a given data format in focus. In this paper, we would like to present our idea using the example of spreadsheets.

## 2   A Pattern Language and Generator for Spreadsheets

Spreadsheets are widely used, especially in the industrial sector. They can model complex workbooks containing multiple sheets with meta data rich cells (content and appearance). We observed several of such workbooks in industry projects and learned from their authors why they modeled and designed spreadsheets in certain ways. From that, we derived for now 11 distinct patterns which form a pattern language for spreadsheets[4]. The structure of these patterns are heavily inspired by Alexander's patterns in the architectural domain [1], however with the difference that our patterns describe for given circumstances (situations, issues, facts) concrete possibilities how to model them using spreadsheets. Typically, such a pattern comes with a title and a context hinting to a specific issue. This circumstance is described in more detail followed by a solution how to store the containing facts in a spreadsheet. After that, an example illustrates with an image how the pattern could be applied. Some patterns provide links to related ones and they are grouped in categories.

On the basis of the proposed pattern language, a data generator called "Data Sprout" was implemented[5]. An online demo[6] lets users generate diverse Microsoft Excel spreadsheets with desired generation patterns from a given RDF graph. Parts of the graph will be differently represented in sheets, depending what generation patterns are activated. Patterns introduce noise in data by uniformly

---

[4] `http://www.dfki.uni-kl.de/~mschroeder/pattern-language-spreadsheets`

[5] `https://github.com/mschroeder-github/datasprout`

[6] `http://www.dfki.uni-kl.de/~mschroeder/demo/datasprout`

or randomly pick different options for each cell, sheet or workbook. In the following, we will describe how Data Sprout produces spreadsheets in general and what altering effects the implemented patterns have (mentioned in italic).

**Layout.** Broadly, the graph's terminology (i.e. classes and properties) determines how sheets are structured while assertions are used to populate cells with data. In Data Sprout's default configuration, each RDF class corresponds to a sheet, whereas each class property (i.e. properties having a domain of that class) corresponds to a column in the sheet. Class instances (i.e. resources which are type of that class) are listed per row, meaning that at respective property columns their objects (resources or literals) are mentioned in cells. The *Multiple Entities in one Cell* pattern allows that multiple objects are listed in a single cell which is useful since RDF graphs are usually multi-edged graphs (i.e. one property has multiple objects). Some generation patterns change the default behavior for structuring tables: *Multiple Types in a Table* makes sure that some randomly picked sheets correspond to two classes. This way, we naturally find instances of different types in one table. *Intra-Cell Additional Information* ensures that some randomly selected columns are related to two or three properties. This means that a single cell can contain multiple RDF nodes (resources and/or literals) from different properties.

**Modelling.** Storing a literal value in a cell can be done in more than one way. Usually, spreadsheets provide in guidelines an intended way, for example, that dates should be stored as numeric values. However, the pattern *Numeric Information as Text* also allows that some literals will be stored using a textual representation instead. In case of resources, the generator has to decide how to mention them in cells, usually by using their labels. Instead of naming them consistently, the pattern *Multiple Surface Forms* ensures that different labeling variations are considered (see [4] in case of persons). Additionally, the generation pattern *Acronyms or Symbols* also ensures the use of short acronym labels and that symbols may represent Boolean literals (e.g. "✓" stands for *true*).

**Formatting.** Instead of a cell's content, its formatting can also convey meaning. Background and foreground color (more precisely: a cell's font color) can be used to encode certain property values, as described in *Property Value as Color*. To carry out this pattern, our generator searches for appropriate property-value-pairs and randomly assigns colors to them. Because colors alone now provide enough information, the corresponding property columns are removed from the sheet. Additionally, spreadsheets allow to format individual parts in texts by using rich text. In case of the *Outdated is Formatted* pattern, our generator ensures that once preselected properties are involved that refer to outdated information, mentioned resources will be crossed out. As soon as multiple objects come from different properties and their relations are not distinguishable anymore, the pattern *Partial Formatting Indicates Relations* can be applied. In this case, our generator randomly allocates colors or styles (like bold, italic or underlined) to properties in order to make the property recognizable again.

Since the generator completes every cell in a sheet, it knows the corresponding statements which were involved during the generation. This provenance infor-

mation describes the intended statements that should be rediscovered when analyzing a cell's content. Such ground truth labels are essential for evaluating approaches that construct knowledge graphs from this data.

## 3   Related Work

In former work [2, Section 3.7] we discussed in more detail the problem of missing publicly available Personal Information Model (PIM) dataset and that pseudo desktop collections do not meet our needs. The most related datasets seem to be test cases for KG construction tools, like the ones provided for RDF Mapping Language (RML) approaches[7]. However, such test sets are far too small and simple, since they are made with a testing purpose in mind. That is why data generation might be a reasonable option. However, after investigating several generator approaches (for a recent survey see [3]), we did not find a suitable one that would produce similar datasets we typically deal with in practice.

## 4   Conclusion and Outlook

For evaluating knowledge graph construction results, we propose the idea to synthetically generate appropriate datasets. Based on our observation that knowledge workers show certain behaviors when creating data, an initial catalog of generation patterns in the domain of spreadsheets was derived. A demonstrator was presented that generates from a given knowledge graph diverse sheets by reproducing such patterns.

In the future, we will extend our pattern language with more patterns and investigate how authentic the generated data appears to users. Moreover, we would like to show that evaluations, whether with real or synthetic data, yield similar results.

***Acknowledgements*** This work was funded by BMBF (gr. no. 01IW20007).

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
