# OpenReview forum: "Dataset Generation Patterns for Evaluating Knowledge Graph Construction"
_eswc-conferences.org/ESWC/2021/Conference/Poster_and_Demo_Track — ESWC2021 P&D_

### Official Review · AnonReviewer3 · 2021-04-14
**This paper shows a demo named Data Sprout based on dataset generation patterns.**

**Rating:** 6
**Confidence:** 2

**Review:**

This is a demo paper, and I have visited the website. The demo is workable. Technically, it defines several patterns for dataset generation. However, I am somehow confused about the relation with knowledge graph. It seems that it only uses some RDF vocabulary elements to describe a spreadsheet.

**Anonymity:**

Yes, I would like my review to remain anonymous.

---

### Official Review · AnonReviewer1 · 2021-04-14
**unclear motivation and work**

**Rating:** 4
**Confidence:** 1

**Review:**

I admit I’m not able to properly evaluate the paper.

quality and clarity.
- The motivation is unclear to me. If the problem is anonimysation, why synthetical data should solve the problem? And exactly, what is the problem? Authors mention “to label such data with intended knowledge graphs to conduct meaningful evaluations” which to me is unclear (label data with what? what are intended kg? evaluations of what?)
- authors propose a pattern language that is not described, neither here nor in prior publications, which makes it hard to evaluate. I read the online documentation (footnote 4 and 5), but no further insights are provided

originality.
despite the motivation and the goals are unclear, the extraction of patterns from tables can be of interest when transforming data into RDF and viceversa.

If I understood correctly, patterns were defined a priori by the developer so as to feed the algorithm “data sprout” which then seems to perform a generic decision tree in the data generation process. I’d have expected the benefits of this work were:
    - patterns semi-automatically detected
    - the generation process does not allow the user to select “the noise” to be introduced, but leverages a more sophisticated way to compare similar data and reproduce the patterns.

significance
- I’m not able to judge. Therefore I'm not confident this work deserves to be presented as-is

**Anonymity:**

Yes, I would like my review to remain anonymous.

---

### Official Review · ~Jose_Emilio_Labra_Gayo2 · 2021-04-14
**Dataset Generation Patterns for Evaluating Knowledge Graph Construction**

**Rating:** 7
**Confidence:** 3

**Review:**

This demo presents a system that generates Excel spreadsheets from RDF data according to some patterns. The generated tables are trying to mimic the patterns created by humans.
I checked the demo and it seems to work as expected. I also reviewed the source code and it is also available in github. The source code seems to contain the list of Turtle files hardcoded (I was expecting to be able to upload any Turtle file) but anyway, it seems that it could work with other Turtle files as well.
The pattern language proposed: http://www.dfki.uni-kl.de/~mschroeder/pattern-language-spreadsheets/ is quite interesting and I think it lacks a bit of background. How are those patterns obtained/named? It seems that the authors gave more importance to the demo and the source code, but those patterns probably deserve some document or paper describing the methodology employed to obtain them. I found them also very well documented. What I am not sure is if that pattern language is exhaustive or not.

Some minor comments:
- The sentence "describe for given circumstances (situations, issues, facts) concrete possibilities how to model them using spreadsheets..." seems a bit strange for me. Maybe adding a "some" and an "about" would make it more readable, i.e. "describe for some given circumstances (situations, issues, facts) concrete possibilities about how to model them using spreadsheets.
- "Patterns introduce noise in data by uniformly or randomly pick<ing> different options..."
- The sentence: "Usually, spreadsheets provide in guidelines an intended way..." is not very clear, is it right?
- The sentence: "...ensures that different labeling variations are considered (see [4] in case of persons)." is also not very clear for me....why is it different for persons?
-


**Anonymity:**

No, I would like my review to be deanonymized.

---

### Official Review · AnonReviewer2 · 2021-04-15
**Interesting work but detail on pattern creation is missing**

**Rating:** 7
**Confidence:** 3

**Review:**

The paper presents a method for generating synthetic data using KGs as a data source that adheres to patterns found in real spreadsheet data. The problem the paper addresses is very relevant and the online demo works very well.
The work has two main components: the patterns and the generator. The explanation of patterns is conflated with the generator, which made it more difficult for me as a reader to understand the patterns and how they were created. I had to follow the link to the patterns to really understand what was being done.

While I understand this is a demo, I believe the short paper should provide a global view of the work, and so suggest to the authors that they add a table describing at least some of the patterns with a simple example for each.

Additionally, a description of how the patterns were captured is missing. This clearly important point is only described by “We observed several of such workbooks in industry projects and learned from their authors why they modeled and designed spreadsheets in certain ways.“ Which industries/domains? Are the patterns generalizable across domains if more than one was addressed? How many workbooks were analyzed, from how many different sources, and how many authors? How was the information collected from authors? Interviews, questionnaires? How was the workbook processing made? All manual, some automated aspects?

A description of the methodology used for creating the patterns would help the reader better understand their potential quality.

Some space can be gained by merging the related work with the introduction, and by removing the descriptions of the patterns that are now mingled in the generator description.


**Anonymity:**

Yes, I would like my review to remain anonymous.

---

### Decision · Program_Chairs · 2021-04-19

Accept